# Sleep Quality Worsens While Perceived Stress Improves in Healthcare Workers over Two Years during the COVID-19 Pandemic: Results of a Longitudinal Study

**DOI:** 10.3390/healthcare10081588

**Published:** 2022-08-22

**Authors:** Haitham Jahrami, Eman A. Haji, Zahra Q. Saif, Noora O. Aljeeran, Aysha I. Aljawder, Faisal N. Shehabdin, Feten Fekih-Romdhane, Khaled Trabelsi, Ahmed S. BaHammam, Michael V. Vitiello

**Affiliations:** 1Department of Psychiatry, Ministry of Health, Manama 410, Bahrain; 2Department of Psychiatry, College of Medicine and Medical Sciences, Arabian Gulf University, Manama 323, Bahrain; 3Department of Psychiatry, The Tunisian Center of Early Intervention in Psychosis, Psychiatry Department “Ibn Omrane”, Tunis 2010, Tunisia; 4Department of Psychiatry, Faculty of Medicine of Tunis, Tunis El Manar University, Tunis 1068, Tunisia; 5High Institute of Sport and Physical Education of Sfax, University of Sfax, Sfax 3000, Tunisia; 6Department of Medicine, College of Medicine, University Sleep Disorders Center, King Saud University, Riyadh 11324, Saudi Arabia; 7The Strategic Technologies Program of the National Plan for Sciences and Technology and Innovation in the Kingdom of Saudi Arabia, Riyadh 11324, Saudi Arabia; 8Department of Psychiatry & Behavioral Sciences, Gerontology & Geriatric Medicine, and Biobehavioral Nursing, University of Washington, Seattle, WA 98195-6560, USA

**Keywords:** anxiety, COVID-19, healthcare workers, PTSD, sleep, stress

## Abstract

The purpose of the study was to measure changes in sleep quality and perceived stress and their interrelationships in a sample of healthcare workers two years post the COVID-19 pandemic. Using a cohort design, data were collected from frontline healthcare workers (FLHCW, *n* = 70) and non-frontline healthcare workers (NFLHCW, *n* = 74) in April 2020 (T1) and in February 2022 (T2). The Pittsburgh Sleep Quality Index (PSQI) and the Perceived Stress Scale (PSS-10) were administered at both time points. There were no differences in sleep quality or perceived stress between FLHCW and NFLHCW at either timepoint. For the entire sample, the PSQI scores at T2 were significantly higher than at T1 (7.56 ± 3.26 and 7.25 ± 3.29, respectively) (*p* = 0.03, Cohen’s d = 0.18). PSS-10 scores at T2 were significantly lower than at T1 (19.85 ± 7.73 and 21.13 ± 7.41, respectively) (*p* = 0.001, Cohen’s d = 0.78). Baseline sleep quality PSQI (T1) was a significant predictor for changes in sleep quality. During the initial months of the outbreak of the COVID-19 pandemic, poor sleep quality and perceived stress were common for healthcare workers. Two years into the pandemic, the perceived stress was reduced, but sleep quality worsened.

## 1. Introduction

The sleep of healthcare workers during the COVID-19 pandemic has been examined in numerous studies worldwide [1,2,3]. Specifically, until July 2021, at least ninety studies had investigated sleep issues in healthcare workers [1]. Previous research showed that during the COVID-19 pandemic, approximately 50% of healthcare workers reported poor sleep, and approximately 30% reported symptoms of insomnia [1]. A recent meta-regression of ninety studies involving about sixty-four thousand healthcare workers investigated the correlates of poor sleep quality in healthcare workers [1]. The results showed that poor sleep quality did not differ by sex, job category (i.e., nurses and physicians) or working in frontline care; however, older age was a statistically significant marker for poor sleep quality [1].

To date, we know that the prevalence of poor sleep quality in healthcare workers during the COVID-19 pandemic is very high. We also know that the prevalence of poor sleep quality in healthcare workers was high before COVID-19. Given healthcare workers’ increased professional and personal duties during the COVID-19 pandemic, sleep loss or poor sleep quality may be exacerbated by added stress over time [3]. However, our knowledge of the sleep quality of healthcare workers during the pandemic comes mainly from cross-sectional studies. A major limitation of the cross-sectional design is that it cannot be used to infer causality because it does not determine the temporal link between the outcome and the exposure, as both are examined at the same time.

Several cross-sectional studies have shown that healthcare workers experienced a considerable escalation in perceived stress as a result of the COVID-19 pandemic [4,5,6]. Results of a systematic review and meta-analysis of mental health problems during the early stages of the COVID-19 pandemic showed that approximately one in every three healthcare workers had experienced moderate–severe stress [7]. The same review concluded that stress and sleep disturbances were the most common complaints among healthcare workers, accounting for 33% and 37%, respectively [7]. The commonly identified work stressors were: workflow disruptions, increased workloads, increased time constraints and fear of contracting or passing the infection to family members [4,5,6,7].

Using a cohort study design, this study’s primary aim was to measure changes in sleep quality and perceived stress in a sample of healthcare workers using validated instruments at the beginning of the pandemic and two years later. A secondary aim was to examine the strength and direction of association between selected demographic and professional characteristics of healthcare workers and their reported change in sleep quality and perceived stress.

We hypothesized that healthcare workers would report improvements in both sleep quality and perceived stress from the initial assessment to follow-up. We expected this result, as people were generally adjusting to the “new normal” in which vaccinations have become widely available, candidate treatments are being administered on an emergency basis, and mortality rates have decreased.

## 2. Materials and Methods

The principles of the Strengthening the Reporting of Observational Studies in Epidemiology (STROBE) recommendations were used to develop and complete this prospective cohort study [8].

### 2.1. Consideration of Ethical Issues

The research procedure was revised and permitted by the Research Ethics Committee of the Ministry of Health, Government of Bahrain (REC/06/04/2020). The research was carried out in accordance with the Declaration of Helsinki for human subject research. Data collection began after the approval. Each participant provided electronic informed consent. The contribution was entirely voluntary, with no financial or other incentives offered, and participants were free to discontinue at any time.

### 2.2. Participants and Setting

Data were collected at two data points: T1 = at baseline, i.e., April 2020 during the early months of the outbreak of the COVID-19 pandemic, and T2 = post emerging COVID-19 pandemic, i.e., February 2022, two years into the pandemic. Both data points used the same online survey to collect data for the study. During T1, Bahrain was under partial lockdown, but this was not the case during T2.

Purposive, non-probability sampling was used to recruit frontline healthcare professionals (FLHCWs, *n* = 70) from two COVID-19 sites run by Bahrain’s Ministry of Health. FLHCWs work in isolation units, fever clinics, nose swab testing clinics and care for patients who have received a COVID-19 diagnosis. Non-frontline healthcare professionals (NFLHCWs, *n* = 74) were chosen, utilizing a similar sampling method from the study team’s acquaintances. Survey participants from both groups were invited to participate through a web link delivered through WhatsApp Messenger.

Participants were required to meet the following criteria: (1) have served in Bahrain’s Ministry of Health for more than six months; (2) be registered healthcare workers, such as physicians, nurses and allied health professionals; and (3) be willing to participate. The study excluded participants who were on leave of absence at the time of the survey or whose data were incomplete at either timepoint.

We estimated that a minimum of 67 pairs are needed to achieve a power of 80% and a level of significance of 5% (two-sided) for detecting an effect size of 0.35 between pairs using paired-sample t-test.

### 2.3. Techniques and Measurements

An English language, self-administered questionnaire was used to collect the data. English is the professional communication language of the practice of healthcare in Bahrain. The survey was composed of structured, close-ended questions and was simple and quick to complete, taking 7–10 min based on pilot testing. The survey comprised basic demographic and professional information: the Pittsburgh Sleep Quality Index (PSQI) [9] and the Perceived Stress Scale (PSS-10) [10].

The demographic and professional data collected included: age (in years), sex (male vs. female), marital status (single vs. married), profession (medical doctor, nurse or other healthcare workers) and type of facility affiliated with (to distinguish between FLHCW and NFLHCW). The PSQI is a well-known self-reported instrument for evaluating sleep quality over a month. It takes 5–10 min to complete the measure, consisting of 19 distinct items that generate seven components that result in a single global score [9]. Subjective sleep quality, sleep latency, sleep length, habitual sleep efficiency, sleep disruptions, usage of sleep-promoting drugs and daily dysfunction owing to drowsiness are the seven components [9]. The PSQI has been utilized in various settings and with a variety of clinical and general populations [11]. According to a recent review, the PSQI has excellent content validity, construct validity and discriminant validity [12]. A cut-off score of five points has a sensitivity of 89.6% and specificity of 86.5% for identifying individuals with poor sleep quality [13]. A global PSQI score of five or greater indicates poor sleep quality [9]. The PSQI global score can range from 0 to 21 points.

The PSS-10, often known as Cohen’s stress scale, is the most commonly used psychological tool for assessing perceived stress [14]. The measure originally used a 14-item scale [15] but was later reduced using factor analysis to a 10-item scale [10] to assess the degree to which stressful occurrences in one’s life are assessed. PSS-10 scores can range from 0 to 40, with higher values suggesting greater perceived stress. Low perceived stress is defined as a score between 0 and 13, moderate stress is defined as a score between 14 and 26, and high perceived stress is defined as a score between 27 and 40 [16]. The PSS-10 is an easy-to-use tool with demonstrated good psychometric features [17].

### 2.4. Statistical Analysis

The data were represented graphically using histograms and box plots before the data analysis began to ensure that they were normal and to spot any potential outliers. To officially assess whether the research variables were normal, the Shapiro–Wilk test was used. Descriptive statistics were used to summarize the findings. Continuous variables were given an arithmetic mean (x¯) and standard deviation (SD), while categorical variables were given counts and percentages. Rates of poor sleep quality and perceived stress were reported; poor sleep quality was defined as PSQI ≥ 5 [18] but was also reported at PSQI ≥ 6 and PSQI ≥ 7 to allow comparison with other studies. For example, the cut-off value for poor sleep quality is ≥7 in Chinese studies [19,20]. Low stress was defined as PSS-10 ≤ 13, moderate stress PSS-10 ≥ 14, and ≤26, high stress PSS-10 ≥ 27 [16].

To compare the two independent groups, the independent sample *t*-test was used for continuous variables, and Chi-square χ2 was used for categorical variables. The paired-samples *t*-test was used to compare the means of two measurements taken from the same individual at T1 and T2. The effect sizes were calculated along with the *p*-value. Regarding the effect size calculations, for categorical variables, Cramer’s V effect sizes were provided, and for continuous variables, Cohen’s d. The effect sizes of Cramer’s V were classified as small effect size 0.10, medium effect size 0.30 and large effect size 0.50 [21]. Cohen’s d was classified as small effect size 0.25, medium effect size 0.50 and large effect size 0.80 [22].

To assess the predictors of change in sleep quality scores and perceived stress, multivariate linear regression was used as dependent variables in separate models. Independent variables included in the model were: age, PSQI (T1), PSS-10 (T1), sex, marital status, professional background and frontline work status. For categorical variables, including dichotomous variables, an indicator of dummy variable (or series of variables) was created to facilitate the interpretation of results.

Linear regression analysis was also used to identify the strength and direction of association between changes in PSS-10 scores as a dependent variable and changes in the seven PSQI components as independent variables. Two models were investigated. In model 1, a simple univariate regression analysis examined the association between changes in PSS-10 scores as a dependent variable and changes in any of the seven PSQI components. In model 2, a multivariate regression analysis examined the association between changes in PSS-10 scores as a dependent variable and changes in the seven PSQI components controlling for the remaining six PSQI components in each subsequent analysis; for example, in the first PSQI component “subjective sleep quality”, the analysis controlled for remaining PSQI items, i.e., “sleep latency”, “sleep duration”, “sleep efficiency”, “sleep disturbance”, “use of sleep medication” and “daytime dysfunction”.

In all statistical modeling, change was assigned the symbol delta (Δ) and was defined as T2–T1, whereby T1 is the “baseline” = initial COVID-19 outbreak, and T2 is the “post” = after two years of the initial COVID-19 outbreak.

In our regression studies, we provided unstandardized coefficients, which are referred to as “raw” coefficients obtained by regression analysis when the study is conducted on the original, unstandardized variables [23]. An unstandardized coefficient has units and a scale that is more like the “real world” than standardized coefficients, which are normalized unit-less coefficients [23].

In the case of multiple regression analyses, the variance inflation factor (VIF) [23], which evaluates the inflation in the variances of the parameter estimates owing to multicollinearity possibly produced by the correlated predictors, was used to quantify multicollinearity [23]. Furthermore, we used tolerance to quantify the extent to which the inclusion of additional predictor variables in a model alters the beta coefficients [24]. Higher degrees of multicollinearity are indicated by smaller amounts of tolerance [24].

The data were analyzed using Stata 17 (version 17 for Windows, 2021, College Station, TX; StataCorp LLC, TX 77845, United States) [25]. In all analyses, a *p*-value < 0.05 was considered statistically significant.

## 3. Results

At T1, 264 of the 280 solicited healthcare professionals participated in the study, representing a response rate of 94%. At T2, a total of 144 healthcare workers responded to all items of the study, representing a response rate of 51% of the original sample. It is worth mentioning here that during T2, because of improvements in COVID-19 in Bahrain, many healthcare workers left COVID-19 treatment sites and returned to their normal clinical work and thus were not eligible to participate in the follow-up. The ratio of FLHCW: NFLHCW was 1:1, with 70 FLHCW and 74 NFLHCW. The mean age for the entire sample was 39.44 ± 9.24, and about 70% were female. The two groups were very homogenous in their sociodemographic and professional characteristics. About 92% were married. Nurses constituted 49%, physicians about 29%, and 22% were other healthcare workers. Table 1 shows the sociodemographic characteristics of the study participants as a whole and a comparison between FLHCW and NFLHCW.

Table 2 provides descriptive results of the study participants regarding the PSQI and PSS. For the entire sample at T1, poor sleep quality was highly prevalent, with PSQI ≥ 5, PSQI ≥ 6 and PSQI ≥ 7 being 76%, 67% and 54%, respectively.

For the entire sample at T2, poor sleep quality was more prevalent, with PSQI ≥ 5, PSQI ≥ 6 and PSQI ≥ 7 being 81%, 72% and 59%, respectively. For the entire sample at T2, the increased rate of poor sleep quality was statistically significant (*p* = 0.03) compared to T1.

There were no statistically significant differences between FLHCW and NFLHCW at T1 or T2 in the prevalence of poor sleep quality (see Table 2).

Regarding the perceived stress for the entire sample at T1, low, moderate and severe stress were 15%, 63% and 22%, respectively. For the entire sample at T2, low, moderate and severe stress were 22%, 58% and 20%, respectively. There were no statistically significant differences between FLHCW and NFLHCW at T1 or T2 in the prevalence of perceived stress (see Table 2).

Table 3 shows that for the entire sample, the sleep quality score at T2 was higher than at T1, with scores of 7.56 ± 3.26 and 7.25 ± 3.29, respectively. The difference of score of 0.31 [−0.58–−0.03] was statistically significant (*p* = 0.03) with small effect size (Cohen’s d = 0.18). The perceived stress scores at T2 were lower than in T1, with scores of 19.85 ± 7.73 and 21.13 ± 7.41, respectively (Table 2). The difference of score of −1.28 [1.01–1.55] was statistically significant (*p* = 0.001) with large effect size (Cohen’s d = 0.78). Two PSQI components worsened (subjective sleep quality and sleep disturbance); four PSQI components improved (sleep latency, sleep duration, sleep efficiency and use of sleep medication); and one PSQI remained unchanged (daytime dysfunction). A statistically significant difference was only observed in the component “use of sleep medication” (see Appendix A).

Table 4 provides the results of the association between selected predictive variables and the change in sleep quality scores and stress scores of the study participants. PSQI at T1 was the single statistically significant predictor of the change in PSQI scores (β = −0.13 *p* = 0.003). Age, PSQI (T1), PSS-10 (T2), sex, marital status, professional background and frontline status did not predict the changes in PSS-10 scores.

Table 5 reports the results of the regression analysis, displaying the association between changes in PSS-10 scores as a dependent variable and changes in the seven PSQI components as independent variables. Results showed that two PSQI components predicted change in PSS-10. Specifically, multivariate regression analysis showed that changes in sleep duration (β = 0.18, *p* = 0.03) and changes in daytime dysfunction (β = −0.17, *p* = 0.04) predicted changes in PSS-10.

Assumption checks for multiple regression analyses showed high tolerance for all variables > 0.94 and a VIF of >1, suggesting that multicollinearity was not an issue.

## 4. Discussion

Healthcare workers experienced high rates of poor sleep quality and perceived stress during the early months of the COVID-19 pandemic. Our T1 data suggest elevated poor sleep and perceived stress were high. Stress levels improved two years into the pandemic, but sleep quality was worse compared to baseline.

In this study, we focused on healthcare workers because, even in normal conditions (i.e., before COVID-19), they have persistently reported concerns about sleep quality. For example, the estimated cumulative prevalence of poor sleep quality in 39 studies with relevant data from nurses was approximately 60% [26]. Additionally, high prevalence rates of poor sleep quality were also reported for physicians in several countries, e.g., Portugal was about 60% [27], the USA was about 30% [28], and Pakistan was about 40% [29]. It seems that poor sleep quality can be present early in a professional medical career and can be present in more than half of the medical students, according to a meta-analysis of forty-three studies involving about eighteen thousand students from thirteen countries [30]. Furthermore, the American Thoracic Society issued a formal statement in 2015 stating that good sleep quality and quantity are essential for healthy physicians [31].

A recent follow-up study by a team in Iran showed that during COVID-19, sleep quality worsened among healthcare workers. Shift workers were found to have a substantial impact on their PSQI score [32].

Sleep quality at baseline was a significant predictor of sleep quality changes at the end of the study. No variable predicted changes in perceived stress scores. Regression analyses demonstrated that changes in perceived stress scores were associated with changes in some components of the global sleep quality index scores; specifically, they were associated with the component related to subjective sleep quality and the component of daytime dysfunction. These findings suggest that there is a relationship between sleep quality (or at least some of its components) and perceived stress.

The main, novel and somewhat counterintuitive finding of our study is that sleep quality worsened over time in the face of improving pandemic conditions, which were associated with decreased perceived stress. From the baseline to follow-up, that is, early in the pandemic’s course until two years later, and the emergence of effective vaccines and intervention drugs, sleep quality among healthcare professionals might have been expected to improve. The small effect size (<0.2) of “statistically” worsened sleep scores might also be clinically interpreted as no change in sleep quality [33].

The unfavorable effects linked to the pandemic may fluctuate with the emergence of COVID-19 variants and the pandemic’s infection rate waves. After the initial wave, during a period with very low rates of COVID-19 infection, a Norwegian cross-sectional study looked into sleep patterns among nurses [34]. The majority of nurses, 84.2%, described that their sleep duration had not changed since the first wave of the COVID-19 pandemic, while 11.9% said they slept less, and 3.9% reported they slept more [34]. Similarly, 82.4% of nurses said their sleep quality had not changed, while 16.2% said their sleep quality had deteriorated since the initial wave of the pandemic [34]. The majority of nurses said the pandemic did not affect their sleep schedule, while 9.6% said they went to bed later, and 9.0% said they woke up earlier than before the outbreak [34]. This suggests that poor sleep quality may vary depending on how and when the data are obtained, e.g., in our study, T1 was during a national lockdown, while this was not the case for T2. Before the COVID-19 pandemic, healthcare professionals had a high rate of poor sleep quality. In Greece, using cross-sectional data, sleep characteristics were compared throughout the first and second periods of restriction measures due to the COVID-19 pandemic; a total of 1078 questionnaires were assessed (*n* = 963 for the first period and *n* = 115 for the second period) [35]. When compared to regular practices, sleep duration increased during the first lockdown and dropped during the second.

Our study showed that sleep quality worsened over time. It is possible that individuals who began to sleep poorly during COVID-19 might have learned to continue to sleep poorly, that is, they developed from new poor sleepers at the inception of the pandemic to chronic poor sleepers two years into the pandemic. The persistence and severity of poor sleep quality can be explained using a behavioral model of insomnia [36,37]. Predisposing variables, precipitating circumstances and perpetuating factors are the three components of Spielman’s 3P model of chronic insomnia [36,37]. According to the model, acute insomnia becomes chronic and self-perpetuating when it becomes chronic [37]. The three factors interact in this model. According to the stress–diathesis conceptualization, the first two factors that lead to insomnia are predisposing and precipitating factors [37]. Behavioral considerations modulate chronicity in the third factor (the perpetuating factor) [37]. Predisposing variables, according to this concept, may cause the occasional night of poor sleep, but the person sleeps well in general until a precipitating event (such as the loss of a loved one, and in the current scenario, the pandemic and its associated risks) occurs, triggering acute insomnia [37]. Insomnia becomes chronic when unhealthy learned behaviors related to sleep or other perpetuating factors emerge and will persist even if the initial [37] predisposing and precipitating factors diminish or disappear [37]. According to a recent meta-analysis of data from individual participants, the prevalence of sleeplessness symptoms during the COVID-19 pandemic was estimated to have been 53%. Approximately 17% of people reported having clinically serious sleeplessness. According to the breakdown, 3% and 14% of people experienced severe insomnia, respectively [38].

Healthcare workers may be affected by poor sleep, stress and mental health issues that affect their ability to think clearly and make clinical decisions [39,40]. Consequently, there is a higher chance of medical errors being made, which might put patients at unnecessary risk [39,40]. According to a 2020 study, physicians with very high sleep-related impairment are 97% more likely to make clinically significant medical errors [41]. Cognitive performance can be reduced by 25% after one night of insufficient sleep [42]. Sleep loss has been compared with alcohol impairment—19 h of sustained wakefulness is equivalent to 0.05% blood alcohol concentration and 24 h to 0.10% [43]. Sleep improvement is a priority for healthcare workers. Based on a survey of nurses, it was found that up to 92% of the healthcare workers were aware of their need to receive a sleep improvement intervention [44]. Despite this, there is no one-size-fits-all solution when it comes to sleep interventions. The next step is therefore to determine which types of sleep interventions would be most effective for healthcare workers. Among inpatient nurses, mindfulness-based strategies were preferred over cognitive-behavioral therapy for insomnia (CBTi) and sleep hygiene education [45]. Perhaps, rather than changing behavior, mindfulness-based strategies focus on bringing the individual back to the present moment [46]. As a result of decreasing physiological arousal and minimizing psychological factors, such as rumination, meditation practices may improve insomnia symptoms. There is evidence that nurses who report insomnia symptoms have a higher perception of stress, contributing to their preference for mindfulness-based strategies [45].

At the initial outbreak of COVID-19, concerns about the high risk of infection, as well as a lack of faith in the safety precautions in place, could have caused occupational stress [47]. In addition, people who believe they do not have enough infection control knowledge and skills are more likely to be stressed [48]. The acquired knowledge and experience about COVID-19 two years into the pandemic might explain the reasons for reduced stress scores. Results from a new longitudinal study support the high incidence of stress symptoms during the COVID-19 pandemic and offer the first longitudinal proof of the impact of felt stress on sleep disruptions during the pandemic [49].

New research revealed that obtaining a COVID-19 vaccine not only protects against infection but can also aid with pandemic-related stress [50]. The study found that immunization had major psychological advantages in addition to lowering the risk of severe illness and death from COVID-19 [50]. By 27 February 2022, Bahrain had administered at least 3,375,361 doses of COVID-19 vaccination. If each person requires two doses, then nearly the entire population could be vaccinated [51].

The association between sleep duration and stress has been studied extensively; most studies have focused on the negative effects of stress on sleeping time [52,53]. Recent research on the general adult population utilized canonical correlation variate analysis to examine the bidirectional relationship between sleep and stress as endogenous variables [54]. Morin and his colleagues provided strong evidence that perceived stress is the main factor in the genesis of insomnia [55].

During the COVID-19 pandemic, individuals with higher anxiety levels were more likely to have higher subjective stress levels and associated poor sleep quality [56]. During the COVID-19 pandemic, recent research of over a thousand people found that the severity of insomnia predicted psychological distress and suicidal ideation [57]. Recent research during the COVID-19 pandemic involving healthcare workers also showed that three out of four healthcare workers experienced poor sleep quality [58]. The same study also reported that shorter sleep duration was linked to a higher rate of psychological distress [58]. The research recommended that sleep could be used as an intervention strategy to reduce psychological stress in healthcare workers [58]. Poor sleep quality has long been known to coexist with mental health issues or function as an indicator of depression and anxiety. Our analyses suggest that interventions are needed to reduce stress and improve sleep quality. Perhaps a practical therapeutic approach for reducing psychological distress among individuals, especially healthcare workers, during the COVID-19 pandemic could be CBTi. Clinical studies of CBTi have shown decreases in anxiety and depression symptoms in patients experiencing both insomnia and psychological distress when used to treat insomnia or shortened sleep duration in non-pandemic conditions [59]. It is possible that treating insomnia is vital because it has been suggested that reduced sleep duration during the COVID-19 pandemic may increase the chance of long-term negative psychological effects [60]. Thus, the prevention and treatment of insomnia, per se, as well as insomnia symptoms in addition to psychological issues should be prioritized. Results of a recent clinical trial during the COVID-19 pandemic concluded that insomnia symptoms could be improved with simplified CBTi, particularly for stress-related acute insomnia [61]. In addition, a task force of the European CBTi academy has made clinical practice guidelines specifically to reduce the burden of stress-related insomnia [62].

### Strengths and Limitations

This study has several advantages that make it novel, for example, the use of a cohort study design to assess the change in sleep quality and perceived stress of healthcare workers during the COVID-19 pandemic. This allowed assessing the temporal association between poor sleep quality/perceived stress and COVID-19, suggesting potential causality. Second, bringing together a diverse group of specialists working both in frontline and non-frontline service allowed for comparisons between different backgrounds.

However, our study has limitations. Our study did not ask about possible confounders, including diets, BMI, medical histories, shift work schedules, pregnancy and concurrent sleep disorders, such as obstructive sleep apnea, in order to make the survey duration reasonable. Objective sleep duration and objective assessment of OSA were beyond the scope of our study, and performing actigraphy or polysomnography was not feasible for clinically active healthcare workers during the COVID-19 pandemic. Other major limitations are the small sample size and reliance on two data points only. Our survey was advertised online; therefore, the results could have been influenced by self-selecting bias. The use of purposive, non-probability sampling was another limitation. Generalizability is limited, as the study focused on clinically active healthcare workers who were directly involved in patient care; however, other healthcare positions, such as administrators and support services, need to be studied. Finally, in the current study, we did not collect data about SARS-COV-2 infection in healthcare workers at baseline and during follow-up. Thus, future research should control for the history of COVID-19 disease, symptoms severity and treatment.

## 5. Conclusions

During the initial months of the outbreak of the COVID-19 pandemic, rates of poor sleep quality and stress were high for healthcare workers. Two years into the pandemic, the stress scores had improved; however, sleep quality worsened. The baseline sleep quality score was a significant predictor for change in sleep quality at the end of the study. Poor sleep quality in healthcare workers linked to the COVID-19 pandemic has the potential to become chronic and long term, suggesting the importance of monitoring and regular follow-up screening of the issue. Preventive programs aimed at reducing perceived stress and improving the sleep quality of healthcare professionals should be implemented during highly stressful events, such as a pandemic.

## Figures and Tables

**Table 1 healthcare-10-01588-t001:** Demographics of the study participants.

Variable *	Total, *n* = 144	FLHCW, *n* = 70	NFLHCW, *n* = 74	*p*-Value **	ES ***
Sex				0.60	0.05
Male	42 (29%)	22 (31%)	20 (27%)		
Female	102 (71%)	48 (69%)	54 (73%)		
Marital status				0.20	0.11
Single	12 (8%)	8 (11%)	4 (5%)		
Married	132 (92%)	62 (89%)	70 (95%)		
Professional background				0.60	0.09
Other HCWs	32 (22%)	17 (24%)	15 (20%)		
Nurses	70 (49%)	31 (44%)	39 (53%)		
Physicians	42 (29%)	22 (32%)	20 (27%)		
Age (years)	39.44 ± 9.24	39.46 ± 9.00	39.42 ± 9.52	0.98	−0.01

Notes: * Frequency count and (%) OR Mean ± SD. ** Independent samples *t*-test or Pearson’s Chi2, significant at *p* < 0.05. *** ES = effect size for categorical variables, Cramer’s V effect sizes were provided, and for continuous variables, Cohen’s d. FLHCW = frontline healthcare workers; NFLHCW = non-frontline healthcare workers; HCW = healthcare worker; PSQI = Pittsburgh sleep quality index; PSS-10 = perceived stress scale; T1 is baseline = initial COVID-19 outbreak; T2 is post = after two years of the initial COVID-19 outbreak.

**Table 2 healthcare-10-01588-t002:** Descriptive results of the study participants.

Variable *	Total, *n* = 144	FLHCW, *n* = 70	NFLHCW, *n* = 74	*p*-Value **	ES ***
Poor sleep quality (T1)					
PSQI ≥ 5	110 (76%)	57 (81%)	53 (72%)	0.2	0.1
PSQI ≥ 6	96 (67%)	49 (70%)	47 (64%)	0.4	0.07
PSQI ≥ 7	78 (54%)	40 (57%)	38 (51%)	0.5	0.06
Poor sleep quality (T2)					
PSQI ≥ 5	117 (81%)	55 (79%)	62 (84%)	0.4	0.07
PSQI ≥ 6	103 (72%)	48 (69%)	55 (74%)	0.45	0.07
PSQI ≥ 7	85 (59%)	42 (60%)	43 (58%)	0.8	0.02
Stress level (T1)				0.20	0.16
Low	21 (15%)	8 (12%)	13 (18%)		
Moderate	91 (63%)	42 (60%)	49 (66%)		
Severe	32 (22%)	20 (28%)	12 (16%)		
Stress level (T2)				0.35	0.12
Low	32 (22%)	14 (20%)	18 (24%)		
Moderate	84 (58%)	39 (56%)	45 (61%)		
Severe	28 (20%)	17 (24%)	11 (15%)		
PSQI (T1)	7.25 ± 3.29	7.57 ± 3.42	6.95 ± 3.16	0.26	−0.19
PSQI (T2)	7.56 ± 3.26	7.57 ± 3.36	7.54 ± 3.19	0.96	−0.01
PSS-10 (T1)	21.13 ± 7.41	21.86 ± 7.08	20.45 ± 7.70	0.26	−0.19
PSS-10 (T2)	19.85 ± 7.73	20.46 ± 7.41	19.28 ± 8.02	0.36	−0.15

Notes: * Frequency count and (%) OR Mean ± SD. ** Independent samples *t*-test or Pearson’s Chi2, significant at *p* < 0.05. *** ES = effect size for categorical variables, Cramer’s V effect sizes were provided, and for continuous variables, Cohen’s d. FLHCW = frontline healthcare workers; NFLHCW = non-frontline healthcare workers; HCW = healthcare worker; PSQI = Pittsburgh sleep quality index; PSS-10 = perceived stress scale; T1 is baseline = initial COVID-19 outbreak; T2 is post = after two years of the initial COVID-19 outbreak.

**Table 3 healthcare-10-01588-t003:** Changes in sleep quality scores and stress scores of the study participants with difference expressed as post-COVID-19 minus baseline.

Variable *	Baseline (T1)	Post-COVID-19 (T2)	Correlation Coefficient	diff. [95% CI] **	VS-MPR ***	*p*-Value ****	Cohen’s d
PSQI	7.25 ± 3.29	7.56 ± 3.26	r = 0.87, *p* = 0.001	0.31 [−0.58–−0.03]	3.42	**0.03**	0.18
PSS-10	21.13 ± 7.41	19.85 ± 7.73	r = 0.98, *p* = 0.001	−1.28 [1.01–1.55]	29.40	**0.001**	0.78

Notes: * Mean ± SD. ** difference (Δ) is post-baseline. *** Vovk–Sellke Maximum p -Ratio: Based on a two-sided *p*-value, the maximum possible odds in favor of H_1_; over H_0_; equals 1/(-e p log(p)) for p0 ≤ 0.37. **** Paired sample t-test, significant at *p* < 0.05. PSQI = Pittsburgh sleep quality index; PSS-10 = perceived stress scale; T1 is baseline = initial COVID-19 outbreak; T2 is post = after two years of the initial COVID-19 outbreak.

**Table 4 healthcare-10-01588-t004:** Association between selected predictive variables and the change in sleep quality scores and stress scores of the study participants.

	Dependent Variable			
Independent Variables *	Model 1		Model 2	
	Δ PSQI		Δ PSS-10	
	β	*p*-Value **	β	*p*-Value **
Age	0.01	0.45	−0.01	0.61
PSQI (T1)	−0.13	**0.003**	0.01	0.83
PSS-10 (T1)	−0.01	0.71	0.02	0.32
Sex	−0.31	0.31	0.26	0.40
Marital status	0.68	0.17	0.39	0.44
Professional background	−0.05	0.81	−0.02	0.90
Frontline HCWs	−0.48	0.08	−0.24	0.41

Notes: * Multiple linear regression. The dependent variable in model 1 and model 2 are changes in PSQI and changes in PSS-10. The independent variables in both models were: age, PSQI (T1), PSS-10 (T1), sex, marital status, professional background and frontline status. ** Significant at *p* < 0.05. HCW = healthcare worker; PSQI = Pittsburgh sleep quality index; PSS-10 = perceived stress scale; T1 is baseline = initial COVID-19 outbreak; T2 is post = after two years of the initial COVID-19 outbreak. Δ = change is defined as T2–T1. β = unstandardized betas.

**Table 5 healthcare-10-01588-t005:** Results of regression analysis displaying the association between changes in PSS-10 scores as dependent variable and changes in the seven PSQI components as independent variables.

Dependent Variable is Δ in PSS-10	Model 1 *		Model 2 *	
Independent Variables	β	*p*-Value **	β	*p*-Value **
Δ Subjective sleep quality	−0.02	0.79	−0.04	0.63
Δ Sleep latency	−0.04	0.63	−0.07	0.37
Δ Sleep duration	0.17	**0.04**	0.18	**0.03**
Δ Sleep efficiency	0.16	0.06	0.16	0.06
Δ Sleep disturbance	−0.04	0.67	−0.05	0.58
Δ Use of sleep medication	−0.05	0.52	−0.09	0.31
Δ Daytime dysfunction	−0.14	0.10	−0.17	**0.04**

Notes: * Model 1 univariate regression analysis, Model 2 multivariate regression analysis controlling for the remaining six PSQI components, e.g., for first component model 1 for the item subjective sleep quality controlled for remaining PSQI items, i.e., sleep latency, sleep duration, sleep efficiency, sleep disturbance, use of sleep medication and daytime dysfunction. ** Significant at *p* < 0.05. PSQI = Pittsburgh sleep quality index; PSS-10 = perceived stress scale. Δ = change is defined as T2–T1. T1 is baseline = initial COVID-19 outbreak; T2 is post = after two years of the initial COVID-19 outbreak. β = unstandardized betas.

## Data Availability

Data can be obtained from the corresponding author based upon request.

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
