# Peer review of "Sleep Quality Worsens While Perceived Stress Improves in Healthcare Workers over Two Years during the COVID-19 Pandemic: Results of a Longitudinal Study"

_healthcare, 2022, doi:10.3390/healthcare10081588_

Round 1
Reviewer 1 Report
In this study, the authors have measured changes in the sleep quality and perceived stress in healthcare workers before and two years post the COVID-19 pandemic. While the concept of this manuscript is important, several limitations need to be addressed prior to consideration of publication.
Major comments
- Introduction is too lengthy and should be presented more compact to the reader. For example the second paragraph could be incorporated in the discussion section.
- Please report the response rate of the participants.
- Additional details are welcome to provide better description of the protocol design. For example did the authors include patients with sleep disorders such as obstructive sleep apnea (OSA), pregnancy or severe comorbidities?
- Did the authors consider other factors such as BMI, excessive daytime sleepiness or co-morbidities as potential explanatory factors in their analysis? Please explain.
- The description of the statistical analysis should be improved; crucial data is missing such as tests of normality.
- Broader discussion is warranted. The authors should discuss more about the clinical implications of their findings, as well as the limitations. For example, recognizing poor sleep quality and its association with patient related outcomes remains a major challenge and the education of clinical healthcare staff and providers about the value and potential positive impact of sleep quality assessment could potentially affect clinical practice.
- Furthermore, a major weakness of the study is the lack of objective indication of sleep duration and the lack of objective (or even questionnaire based) exclusion of OSA diagnosis.
- Another Important limitation is that, because of the open invitation for participation in the study and the lack of randomization of patients, patients with more frequent symptoms and greater concern about poor sleep quality may have accepted the invitation, which would explain the relatively high prevalence of poor sleep quality in the sample.
Author Response
In this study, the authors have measured changes in the sleep quality and perceived stress in healthcare workers before and two years post the COVID-19 pandemic. While the concept of this manuscript is important, several limitations need to be addressed prior to consideration of publication.
Authors’ response: Dear Reviewer, we have addressed all concerns raised, as highlighted below. Changes and additions to the manuscript in response to these concerns appear in red font for convenience of review in both letter and manuscript.
Major comments
- Introduction is too lengthy and should be presented more compact to the reader. For example, the second paragraph could be incorporated in the discussion section.
Authors’ response: Paragraph number 2 of the introduction was moved to the discussion to ensure that introduction is compact.
- Please report the response rate of the participants.
Authors’ response: We added the following paragraph: "At T1, 264 of the 280 solicited healthcare professionals participated in the study, representing a response rate of 94%. At T2, a total of 144 healthcare workers responded to all items of the study, representing a response rate of 51% of the original sample. It is worth mentioning here that during T2, because of improvements in COVID-19 in Bahrain, many healthcare workers left COVID-19 treatment sites and returned to their normal clinical work and thus were not eligible to participate in follow-up. “
- Additional details are welcome to provide better description of the protocol design. For example did the authors include patients with sleep disorders such as obstructive sleep apnea (OSA), pregnancy or severe comorbidities?
Authors’ response: Our study did not ask about possible confounders including diets, BMI, medical histories, shift work schedules, pregnancy, concurrent sleep disorders such obstructive sleep apnea in order to make survey duration reasonable.
Thus, we added the above to limitations.
The above limitation was acknowledged in our initial data collection back in 2020: Jahrami H, BaHammam AS, AlGahtani H, Ebrahim A, Faris M, AlEid K, Saif Z, Haji E, Dhahi A, Marzooq H, Hubail S, Hasan Z. The examination of sleep quality for frontline healthcare workers during the outbreak of COVID-19. Sleep Breath. 2021 Mar;25(1):503-511. doi: 10.1007/s11325-020-02135-9. Epub 2020 Jun 26. PMID: 32592021; PMCID: PMC7319604.
- Did the authors consider other factors such as BMI, excessive daytime sleepiness or co-morbidities as potential explanatory factors in their analysis? Please explain.
Authors’ response: We did not consider other factors such as BMI, EDS, medical histories, shift work schedules; thus, we added the above to limitations.
- The description of the statistical analysis should be improved; crucial data is missing such as tests of normality.
Authors’ response: We added the following description to the statistical analysis section: “The data were represented graphically using histograms and box plots before the data analysis began to ensure that they were normal and to spot any potential outliers. To officially assess whether the research variables were normal, the Shapiro-Wilk test was used”.
- Broader discussion is warranted. The authors should discuss more about the clinical implications of their findings, as well as the limitations. For example, recognizing poor sleep quality and its association with patient related outcomes remains a major challenge and the education of clinical healthcare staff and providers about the value and potential positive impact of sleep quality assessment could potentially affect clinical practice.
Authors’ response: We have added the following paragraph to the discussion: “Healthcare workers may be affected by poor sleep, stress, and mental health issues that affect their ability to think clearly and make clinical decisions [39,40]. The presence of these factors can result in a greater chance of medical errors being made, putting patients at higher risk [39,40]. According to a 2020 study, physicians with very high sleep-related impairment are 97% more likely to make clinically significant medical errors [41]. Cognitive performance can be reduced by 25% after one night of insufficient sleep [42]. Sleep loss has been compared with alcohol impairment - 19 hours of sustained wakefulness is equivalent to 0.05% blood alcohol concentration, and 24 hours to 0.10% [43]. Sleep improvement should clearly be a priority for healthcare workers. Based on a survey of nurses it was found that up to 92% of these healthcare workers were aware of their need to receive a sleep improvement intervention [44]. Despite this, there is no one-size-fits-all solution when it comes to sleep interventions. The next step is therefore to determine which types of sleep interventions would be most effective for healthcare workers. There is evidence that nurses who report insomnia symptoms have a higher perception of stress, contributing to their preference for mindfulness-based strategies [45]. Among inpatient nurses, mindfulness-based strategies were preferred over cognitive-behavioral therapy for insomnia (CBTi) and sleep hygiene education [45]. Rather than changing behavior, mindfulness-based strategies focus on bringing the individual back to the present moment [46]. As a result of decreasing physiological arousal and minimizing psychological factors like rumination, such practices may improve insomnia symptoms.”
- Furthermore, a major weakness of the study is the lack of objective indication of sleep duration and the lack of objective (or even questionnaire based) exclusion of OSA diagnosis.
Authors’ response: Objective assessment of sleep duration and OSA was beyond the scope of our study and performing actigraphy or polysomnography was not feasible for clinically active healthcare workers during the COVID-19 pandemic. We added the following to the strengths and limitations: “Objective sleep duration and objective assessment of OSA were beyond the scope of our study, and performing actigraphy or polysomnography was not feasible for clinically active healthcare workers during the COVID-19 pandemic”.
The lack of assessment of OSA or other formal sleep disorders using subjective methods was addressed as limitations as described above in points #3, and #4.
- Another Important limitation is that, because of the open invitation for participation in the study and the lack of randomization of patients, patients with more frequent symptoms and greater concern about poor sleep quality may have accepted the invitation, which would explain the relatively high prevalence of poor sleep quality in the sample.
Authors’ response: We agree with the reviewer, and we acknowledged this in our original submission by stating that “Our survey was advertised online; therefore, the results could have been influenced by self-selecting bias. The use of purposive, non-probability sampling was another limitation”.

Reviewer 2 Report
Some of the comments the authors can address to improve the quality of the paper:
1. Line 158 “arithmetic mean (x̄) ”, I recommend to check the font in parentheses.
2. Please unify position of the text in Table 3.
3. Please check the sentence “...and worsening of sleep quality 33,34.” (line 326).
4. There are some grammatical mistakes in the paper that need to be corrected.
5. I noticed that single and double quotation marks are used together, I am wondering about the difference of meaning between them in the paper.
6. If those results of significant differences can be marked in the table would be better.
Author Response
Some of the comments the authors can address to improve the quality of the paper:
Authors’ response: Dear Reviewer, we have addressed all concerns raised, as highlighted below. Changes and additions to the manuscript in response to these concerns appear in red font for convenience of review in both letter and manuscript.
- Line 158 “arithmetic mean (x̄) ”, I recommend to check the font in parentheses.
Authors’ response: We modified the font of the arithmetic mean (x̄) symbol to Ariel in MDPI native Palatino Linotype font the symbol (x̄) looks like Chi2..
- Please unify position of the text in Table 3.
Authors’ response: We unified the position to Align top left in Table 3; we also confirm uniformity with other Tables in the manuscript.
- Please check the sentence “...and worsening of sleep quality 33,34.” (line 326).
Authors’ response: We revised the sentence for clarity and now reads as “The persistence and severity of poor sleep quality can be explained using a behavioral model of insomnia [36,37].”
- There are some grammatical mistakes in the paper that need to be corrected.
Authors’ response: The paper was rechecked for grammar, spelling and punctuation by a senior author who is native English speaker. We will further work with MDPI production team to address any issues during proof.
- I noticed that single and double quotation marks are used together, I am wondering about the difference of meaning between them in the paper.
Authors’ response: The use of “ (double quotation) or ‘ (single quotation) was unsystematic and therefore we unified it to “ (double quotation) throughout the manuscript.
- If those results of significant differences can be marked in the table would be better.
Authors’ response: In all Tables p<0.05 was highlighted as bold text.

Reviewer 3 Report
Dear Authors
The work presented to me for review is interesting and examines current issues. Below I refer to individual elements of the work and indicate the necessary changes.
Article: Sleep quality worsens while perceived stress improves in 2 healthcare workers over two years during the COVID-19 pan- 3 demic: results of a longitudinal study
Title: The title of the article is consistent with the content of the work
Abstract: Abstract presents the content of the work in a factual and consistent manner.
Introduction: The introduction comprehensively describes the theoretical foundations of the research based on well-chosen literature.
Material and methods: The authors describe the material and methods in great detail. Consideration of ethical issues, participants and setting, techniques and measurements, and statistical analysis are included.
Results: The obtained results were presented in a descriptive and graphic manner. The tables (4) are well described and explained.
Discussion: The discussion requires the ordering and verification of the cited literature (see References)
Conclusion: The conclusions are made to the point. The conclusions correspond to the main purpose of the work.
References: Some references appear to be out of date (e.g. 15, 16, 19, 21, 24, 27, 29, 33, 36, 37, 42, 43, 45). I encourage authors to use more recent research.
Yours sincerely,
Reviewer
Author Response
Dear Authors
The work presented to me for review is interesting and examines current issues. Below I refer to individual elements of the work and indicate the necessary changes.
Authors’ response: Dear Reviewer, we have addressed all concerns raised, as highlighted below. Changes and additions to the manuscript in response to these concerns appear in red font for convenience of review in both letter and manuscript. We thank you for the nice comments.
Article: Sleep quality worsens while perceived stress improves in healthcare workers over two years during the COVID-19 pandemic: results of a longitudinal study
Authors’ response: No action was needed.
Title: The title of the article is consistent with the content of the work
Authors’ response: No action was needed.
Abstract: Abstract presents the content of the work in a factual and consistent manner.
Authors’ response: No action was needed.
Introduction: The introduction comprehensively describes the theoretical foundations of the research based on well-chosen literature.
Authors’ response: No action was needed.
Material and methods: The authors describe the material and methods in great detail. Consideration of ethical issues, participants and setting, techniques and measurements, and statistical analysis are included.
Authors’ response: No action was needed.
Results: The obtained results were presented in a descriptive and graphic manner. The tables (4) are well described and explained.
Authors’ response: No action was needed.
Discussion: The discussion requires the ordering and verification of the cited literature (see References)
References: Some references appear to be out of date (e.g., 15, 16, 19, 21, 24, 27, 29, 33, 36, 37, 42, 43, 45). I encourage authors to use more recent research.
Authors’ response: We would like to clarify that references (15, 16, 19, 21, 24, 27, 29, 33, 36, 37) are used appropriately as they describe methodological issues e.g., instruments validation or statistical guidelines or interpretation of findings using landmark models such as insomnia’s 3P model. References (42, 43, 45) are published in 2021-2022.
Because we moved Paragraph 2 to the discussion numbers has changed (15, 16, 19, 21, 24, 27, 29, 33, 36, 37, 42, 43, 45) are now ( 9, 10, 13, 14, 18, 21, 23, 32, 36, 37, 52, 53, 55).
We further added two new references (2022) to enhance discussion. These are:
- AlRasheed, M.M.; Fekih-Romdhane, F.; Jahrami, H.; Pires, G.N.; Saif, Z.; Alenezi, A.F.; Humood, A.; Chen, W.; Dai, H.; Bragazzi, N., et al. The prevalence and severity of insomnia symptoms during COVID-19: A global systematic review and individual participant data meta-analysis. Sleep Medicine 2022, https://doi.org/10.1016/j.sleep.2022.06.020, doi:https://doi.org/10.1016/j.sleep.2022.06.020.
- Ballesio, A.; Zagaria, A.; Musetti, A.; Lenzo, V.; Palagini, L.; Quattropani, M.C.; Vegni, E.; Bonazza, F.; Filosa, M.; Manari, T., et al. Longitudinal associations between stress and sleep disturbances during COVID-19. Stress and Health n/a, doi:https://doi.org/10.1002/smi.3144.
Conclusion: The conclusions are made to the point. The conclusions correspond to the main purpose of the work.
Authors’ response: No action was needed.

Round 2
Reviewer 1 Report
The authors have sufficiently answered to all reviewers comments